# Quantitative Analysis of Climate Variability and Human Activities on Vegetation Variations in the Qilian Mountain National Nature Reserve from 1986 to 2021

Xiaoxian Wang [1,2], Xiuxia Zhang [1,2], Wangping Li [1,2,*], Xiaoqiang Cheng [1,2], Zhaoye Zhou [1,2], Yadong Liu [3,4], Xiaodong Wu [3,4], Junming Hao [1,2], Qing Ling [1,2], Lingzhi Deng [1,2], Xilai Zhang [1,2] and Xiao Ling [5]

1   School of Civil Engineering, Lanzhou University of Technology, Lanzhou 730050, China;
    wangxiaoxian@lut.edu.cn (X.W.); zhangxx@lut.edu.cn (X.Z.); chengxiaoqiang@lut.edu.cn (X.C.);
    zhou_zy@lut.edu.cn (Z.Z.); haojm198@lzb.ac.cn (J.H.); lingq@lut.edu.cn (Q.L.);
    222081600001@lut.edu.cn (L.D.); 222085704008@lut.edu.cn (X.Z.)
2   Gansu Emergency Mapping Engineering Research Center, Lanzhou 730050, China
3   Cryosphere Research Station on the Qinghai-Tibet Plateau, State Key Laboratory of Cryospheric Science,
    Northwest Institute of Eco-Environment and Resource, Chinese Academy of Sciences, Lanzhou 730000, China;
    liuyadong@nieer.ac.cn (Y.L.); wuxd@lzb.ac.cn (X.W.)
4   College of Resources and Environment, University of Chinese Academy Sciences, Beijing 100049, China
5   School of Petrochemical Engineering, Lanzhou University of Technology, Lanzhou 730050, China;
    lingxiao_lut@163.com
*   Correspondence: lwp_136@lut.edu.cn; Tel.: +86-180-9311-1731

**Abstract:** Rapid climate variability and intense human activities generate obvious impacts on the Qilian Mountains ecosystem. The time series of fractional vegetation coverage (FVC) from 1986 to 2021 were used to quantify the impact of climate variability and human activities on vegetation variations in the Qilian Mountain National Nature Reserve (QMNNR), using 3147 land satellite images based on the Google Earth Engine cloud platform. The contributions of climate variability and human activities to FVC were quantified using multiple regression residual analysis. Partial correlation and correlation methods were used to quantify the impact of temperature, precipitation, and human activity footprints on FVC. The results showed that from 1986 to 2021, the increase rate of FVC was $1.7 \times 10^{-3}$ $\text{y}^{-1}$, and the high vegetation coverage of the FVC was mainly distributed in the southeastern part of the reserve. In contrast, the low vegetation coverage was mainly distributed in the northwest part of the reserve. The Mann–Kendall mutation test found that the year of 2009 was the year of the mutation. The growth rate of FVC from 2010 to 2021 was greater than that from 1986 to 2009. In addition, climate variability and human activities exhibited a remarkable spatial heterogeneity in FVC changes. Climate variability and human activities contributed 49% and 51% to the increase in FVC in the reserve, respectively, and the contribution of human activities was greater than that of climate variability. The warming and humidification phenomena in the reserve were obvious. However, precipitation was the dominant factor affecting the dynamic changes in FVC. This study improves our understanding of the response of vegetation dynamics to the climate and human activities in the QMNNR.

**Keywords:** fractional vegetation coverage; analysis of drivers; climate variability; human activities; Qilian Mountain National Nature Reserve

## 1. Introduction

Vegetation is a crucial part of terrestrial ecosystems [1], playing an important ecological role in promoting groundwater recharge [2], the material cycle, and carbon regulation [3,4]. Fractional vegetation coverage (FVC) is an important indicator for monitoring changes in the ecological environment [5] and is also an important parameter reflecting the dynamic

characteristics of vegetation and plays a relevant role in monitoring regional ecological environment quality [6,7]. Recent research has shown that "greening the planet" is prominent in China and India due to the impacts of climate variability and human activities [8]; however, the vegetation in some other regions, such as high latitudes and North America, has gradually stabilized or even decreased [9]. Vegetation changes in the Qinghai–Tibet Plateau are a combined effect of human activities and climate variability [10–12]. In contrast, the distribution and changes in vegetation patterns in the Qinghai–Tibet Plateau are significantly affected by temperature increases and precipitation variation [13,14]. Additionally, human activities, including the implementation of ecological restoration projects [15] and grazing, promote vegetation changes [16]. Therefore, clarifying the impact of climate variability and human activities on vegetation changes can effectively support the formulating of reasonable policies for ecological environment restoration.

Climate variability alters the regional climate by driving vegetation growth [17,18]. Air temperature and precipitation promote the overall greening of global vegetation [19] and are the main factors influencing vegetation growth [20]. Chen [21] showed that different climate conditions in the Tibetan Plateau affect the spatial differences in vegetation response to temperature and precipitation. For example, vegetation in the northern area of the Tibetan Plateau is positively correlated with precipitation, whereas the vegetation in the southern area is positively correlated with temperature. Huang demonstrated [22] that the comprehensive effects of temperature and precipitation on the Qinghai–Tibet Plateau show strong spatial heterogeneity, and precipitation dominates vegetation growth in arid steppe and meadow regions. With global warming, the severely cold climate in the Qinghai–Tibet Plateau region has improved, and the vegetation belt has extended to high altitudes and latitudes [23]. Studies have revealed the impact of changes in the air temperature and precipitation on spatial changes in the FVC; however, few studies have quantified the temperature and precipitation contributions. Therefore, understanding the influence of the climate on regional-scale vegetation is of great significance for the ecologically sustainable development of reserves.

The impact of human activities on vegetation coverage changes can be positive or negative [24,25]. In the past few decades, human activities in the Tibetan Plateau have mainly been based on grazing [26]. With economic development, meadows in the Tibetan Plateau have been seriously degraded due to overgrazing [27]. In contrast, with increasing emphasis on ecological environment protection, the Chinese government has launched a series of ecological restoration projects, including the Three Northern Shelter Forest Program [28], the Grain for Green Program [29], and the Ant Forest Project [30,31]. These projects aim to restore the ecological environment by planting trees and protecting local natural forests. Ecological engineering projects have greatly promoted vegetation restoration in semiarid and subhumid areas in Northern China [32]. However, there are few studies specifically clarifying the impact of human activities on the FVC.

Large-scale vegetation dynamics do not reflect detailed features at small regional scales. Previous studies have mostly retrieved the FVC based on remote sensing vegetation indices, among which the normalized difference vegetation index (NDVI) is the most widely used [33]. Commonly used NDVI data include SPOT-NDVI [34], AVHRR-NDVI [35], and MODIS-NDVI [36]. Although these data have been updated several times, their application in vegetation monitoring is limited due to their low spatial resolution and short time series. The NDVI data obtained using Landsat has a longer time series and better spatial resolution and can be used to monitor vegetation changes in the QMNNR. With the development of remote sensing cloud computing platforms, such as the Google Earth Engine (GEE), researchers can use these cloud platforms to conduct research on large-area, long-term series of high-resolution image data [37,38], thus overcoming the data resolution barriers of restriction. In recent years, various methods have been used to analyze the attribution of climate variability and human activities to vegetation, such as statistical methods [39,40], the partial correlation method [41], and the multiple regression residual analysis method [42]. Statistical methods have high requirements on the completeness and

accuracy of historical statistical data, and a single method cannot fully clarify the causes of vegetation changes [43]. Combining statistical methods and partial correlation analysis methods can clearly quantify the factors affecting vegetation changes [44,45]. Based on the assumption of FVC change in the study area, multiple regression residual analysis can better quantify the relative influence of climate variability and human activities to FVC change [46]. Therefore, statistical methods, first-order correlation analysis methods, and multiple regression analysis methods were widely used to elucidate the effects of human activities and climate on the FVC. At present, it is impossible to quantify specific human contributions, as the data related to human activities may not be fully expressed on a spatial scale. The human activity footprint dataset [47] is generally made via weighted summation of eight variables (including the building environment, population density, night lights, cultivated land, pastures, roads, railways, and navigable waterways) that reflect human pressure. This dataset enables a better understanding of the scope and intensity of human impact.

The Qilian Mountains are located in the northeastern part of the third pole of the earth (the Tibetan Plateau). The Qilian Mountains are a priority area for biodiversity conservation in China, an important water sources for the Yellow River Basin [48], and play a crucial strategic role in maintaining local ecological security [49]. The QMNNR is located in the north of the Qilian Mountains. In recent years, many problems, such as vegetation degradation, glacier melting, and soil erosion, have occurred in this reserve [50]. Previous studies have shown that vegetation changes in the reserve are affected by various factors, mainly including extreme events, such as temperature and precipitation [51], and human activities, such as illegal prospecting, mining, and construction of water conservancy projects [52]. To date, few studies have analyzed the change in FVC in the reserve on a geographical scale. Therefore, research on the dynamic changes in the FVC in reserves at the municipal level is of great significance for further exploring the evolution of ecological environment quality in long-term sequences of the reserve and comprehensively managing the ecological environment of the reserve.

Therefore, this study quantitatively analyzed the temporal and spatial variation characteristics of the FVC in the QMNNR from 1986 to 2021, using the GEE cloud platform to construct a FVC time series using 3147 Landsat remote sensing images. Based on this, temperature, precipitation, and human activity footprint data were introduced to clarify the influence of human activities and climate variability on the FVC. The findings of this study may provide an important reference for the restoration of the ecological environment in the reserve.

## 2. Materials and Methods

### 2.1. Area

The QMNNR (97°23′34″–103°45′49″ N, 36°29′57″–39°43′39″ E) was established in 1988 with the approval of the State Council forest and wildlife type nature reserves (Figure 1). The total area of the reserve is about $2.65 \times 10^4$ km$^2$, and the functional area is divided into a core area ($5.05 \times 10^3$ km$^2$), a buffer zone ($3.87 \times 10^3$ km$^2$), and an experimental area ($1.09 \times 10^4$ km$^2$) [53]. Most of the reserve is situated 3000–3500 m above sea level, and precipitation is mainly distributed from May to September, with the most precipitation concentrated in the period from July to August. The annual average precipitation is between 300 and 500 mm, and the annual average temperature is 1.0–4.0 °C. This reserve has a plateau continental climate, with long and cold winters and short and mild summers. The climate elements change regularly from bottom to top with the elevation of the mountains, showing obvious vertical climate belts [54]. This reserve covers forests, meadows (Figure 1a), rivers, glaciers (Figure 1b), and other ecological resources, with rich and diverse species. For a long time, ecological damage problems, such as local water and soil erosion (Figure 1c), have been very serious in the Qilian Mountains.

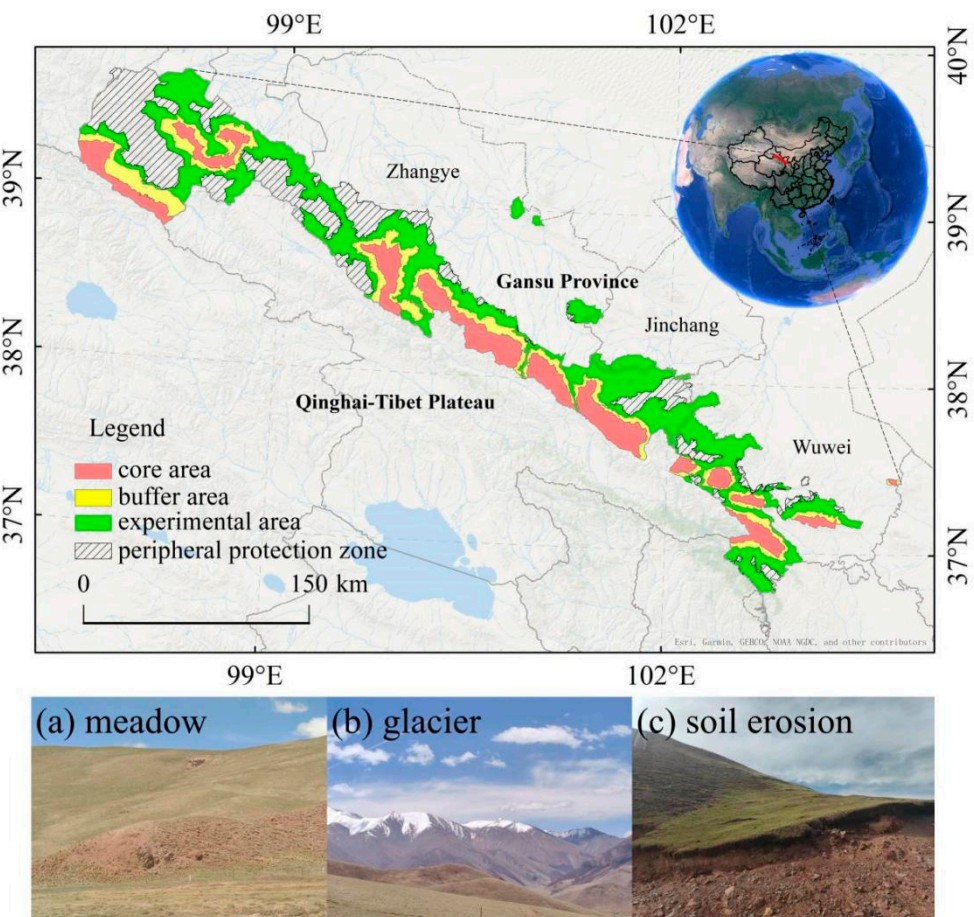

**Figure 1.** Overview of the Qilian Mountain National Nature Reserve.

*2.2. Data Source and Preprocessing*

We selected Landsat 5/7/8 SR data based on the Landsat Collection Tier 1 dataset available online on the GEE platform. Considering the seasonal changes in vegetation [55], low cloud cover (cloud < 20%) screening was performed on the remote sensing images of the reserve from June to September in the years from 1986 to 2021. The NDVI was obtained according to the band operation. The maximum value synthesis algorithm was used to synthesize the NDVI image, and FVC was calculated by combining the dimidiate pixel model. Table 1 lists the data used in the study.

**Table 1.** Sources of data used in this study (All data accessed on 1 October 2022).

| | Dataset | Type | Image Usability Analysis | Spatial Resolution/m | Time Resolution/Year | Data Source |
|---|---|---|---|---|---|---|
| Image data | Landsat 5 SR | Raster | 1326 scenes | 30 | 1986–2011 | United States Geological Survey https://www.usgs.gov/ |
| | Landsat 7 SR | Raster | 1167 scenes | 30 | 1999–2021 | United States Geological Survey https://www.usgs.gov/ |
| | Landsat 8 SR | Raster | 654 scenes | 30 | 2013–2021 | United States Geological Survey https://www.usgs.gov/ |
| Basic data | Landsat path row (WRS–2) | Vector | / | / | 1983–now | Geodata Platform, School of Urban and Environmental Studies, Peking University http://geodata.pku.edu.cn |
| | Product data | Raster | / | 30 | 2019–2021 | National Qinghai–Tibet Plateau Scientific Data Centre https://data.tpdc.ac.cn/zh-hans/ |
| | Temperature and precipitation data | Raster | / | 1000 | 1986–2020 (monthly) | Climatic Research Unit gridded Time Series https://crudata.uea.ac.uk/cru/data/hrg/ |
| | Human footprint dataset | Raster | / | 1000 | 2000–2018 | [47] |

In this study, the availability of Landsat images in the reserve was analyzed based on the GEE cloud platform, with a total of 3147 images. Figure 2 shows the spatial distribution (Figure 2a), temporal distribution (Figure 2b), frequency of images (Figure 2c), and number of available images (Figure 2d) for the period 1986–2021.

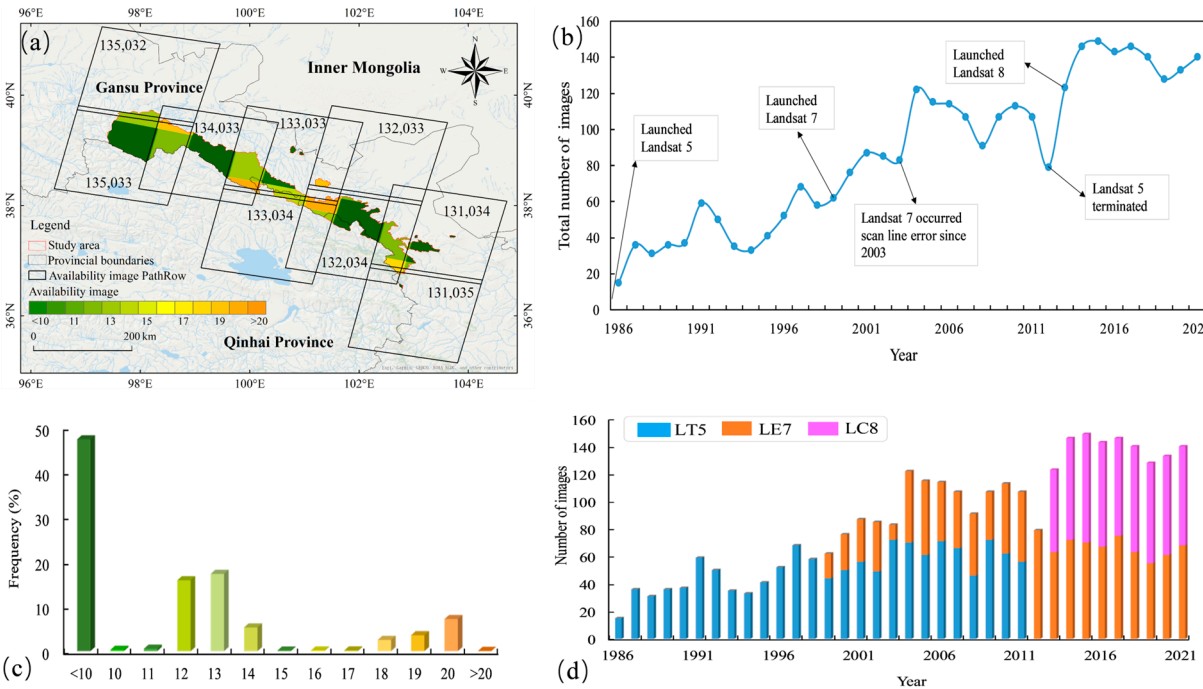

**Figure 2.** Availability of Landsat images of a time–space series of the reserve from 1986 to 2021. (**a**) The range of the Landsat Worldwide Reference System 2 (WRS–2) covering the reserve, (**b**) Landsat image time distribution, (**c**) Landsat image frequencies, and (**d**) total number of sensor images (Landsat 5/7/8).

## 2.3. Methods

An overview of the methodology and structural framework is shown in Figure 3, with the following main steps: (I) Landsat remote sensing image availability analysis and data pre-processing. Based on the GEE platform, the FVC time series was constructed using the maximum value compositing (MVC) [56] combined with the dimidiate pixel model. Climate data, human activity footprint data, and product data were preprocessed using ArcGIS 10.4 software; (II) construction of a dimidiate pixel model and spatial and temporal trend analysis using Theil–Sen analysis and the Mann–Kendall test. Multiple regression residual analysis, correlation, and partial correlation methods were used to explore the driving factors of FVC changes; (III) driving analysis. A binary linear regression model was established based on growing season FVC, temperature, and precipitation. The impact of climate variability and human activities on the FVC was quantified. The effects of temperature and precipitation and human footprint on the FVC were quantified using the correlation and partial correlation methods; (IV) consistency check of the study results. The results of this study were tested for their reliability based on 30 m product data from the Tibetan Plateau Science Data Center.

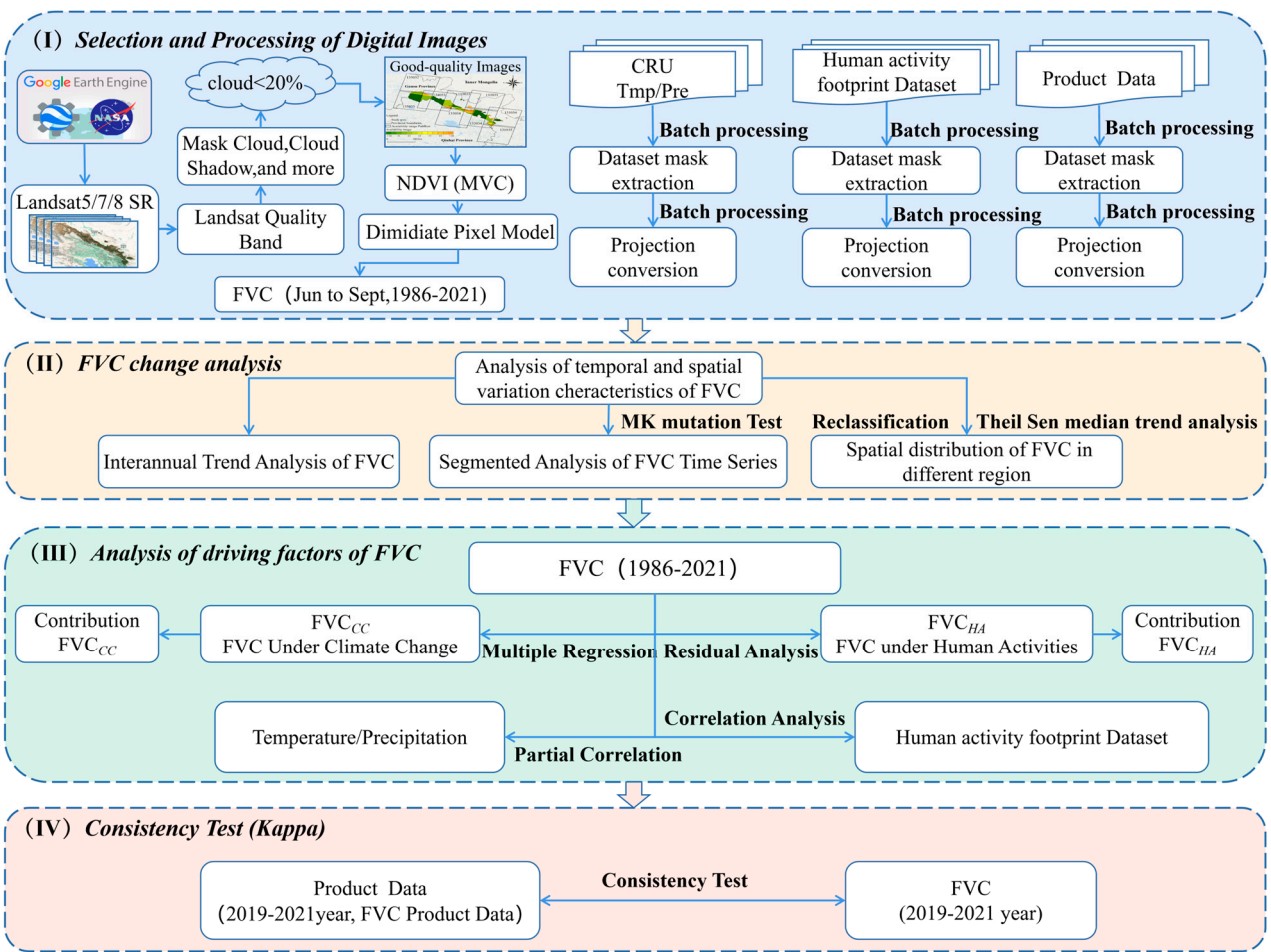

**Figure 3.** Flowchart showing the four main steps of this study. MVC indicates the maximum value compositing. CRU denotes the temperature and precipitation data produced by the UK National Center for Atmospheric Science.

### 2.3.1. Retrieval of Fractional Vegetation Coverage

The normalized vegetation index (*NDVI*), compared with other vegetation indices, has the advantages of high sensitivity and wide-area monitoring for vegetation monitoring, and the ability to eliminate shadows and radiation interference from topography and community structure, as well as the noise from the solar altitude angle and atmosphere. The specific calculation formula used is shown in Formula (1):

$$NDVI = \frac{NIR - \text{Red}}{NIR + Red} \tag{1}$$

where *NIR* is the near-infrared band of the Landsat image, and *Red* denotes the red band of the Landsat image.

The dimidiate pixel model in the mixed pixel model expresses the linear relationship between remote sensing information and vegetation coverage in the same way as the linear regression model, and has the advantages of simplicity, no geographical restrictions, and easy promotion. The dimidiate pixel model also makes up for the shortcomings of the *NDVI* in terms of soil background and atmospheric influence [57]. The FVC is better able to measure surface vegetation conditions and ecological environment changes [58]. The principle of calculating the FVC based on the dimidiate pixel model is to assume that each image has two parts, soil and vegetation, one for the image with vegetation cover (*NDVI*$_{veg}$) and the other for the image without vegetation cover (*NDVI*$_{soil}$), where the value of each

image is a linearly weighted composite of $NDVI_{veg}$ and $NDVI_{soil}$. The specific calculation formula employed is shown in Formula (2):

$$FVC = \frac{NDVI - NDVI_{soil}}{NDVI_{veg} - NDVI_{soil}} \tag{2}$$

where $NDVI_{soil}$ represents the value corresponding to no vegetation or bare soil, and $NDVI_{veg}$ indicates the value corresponding to vegetation.

In theory, $NDVI_{soil}$ is 0 and $NDVI_{veg}$ is 1. However, these data can be affected by necrotic pixels and noise in the remote sensing image itself. Pixels without vegetation and pixels with pure vegetation cannot reach the theoretical value. We referred to previous relevant research results [46] and defined the NDVI value in the NDVI frequency histogram of the image corresponding to a cumulative frequency, with a value of 5% as $NDVI_{soil}$ and a value of 95% as $NDVI_{veg}$. As suggested in a previous study [59], and in terms of the actual situation of the reserve, the FVC was divided into five classes, namely low [0–0.2], medium–low [0.2–0.4], medium [0.4–0.6], medium–high [0.6–0.8], and high [0.8–1.0] vegetation cover.

### 2.3.2. Theil–Sen Median Trend Analysis and the Mann–Kendall Test

Using the combination of Theil–Sen trend analysis and the Mann–Kendall test, the interannual variation trend of FVC and its significance level were discussed [60,61]. Theil–Sen median analysis can effectively eliminate the impact of invalid values by calculating the median of the time series, which is calculated by Equation (3):

$$\beta = median\left(\frac{FVC_j - FVC_i}{j - i}\right), 1986 < i < j < 2021 \tag{3}$$

where $\beta$ denotes the trend of FVC, $median\ (FVC_j - FVC_i/j - i)$, $1986 < i < j < 2021$) is the median function, and $FVC_i$ and $FVC_j$ are the FVC values of the $i$ and $j$ years, respectively. When $\beta > 0$, FVC trends to rise, and vice versa.

### 2.3.3. The Mann–Kendall Nonparametric Test

The Mann–Kendall method was not disturbed by invalid values, and was used to reveal the mutation phenomenon of the FVC trend time series characteristics during the study period [62].

$$S = \sum_{i=1}^{n-1} \sum_{j=i+1}^{n} sign(FVC_j - FVC_i) \tag{4}$$

$$sign(FVC_j - FVC_i) = \begin{cases} 1\ FVC_j - FVC_i > 0 \\ 0\ FVC_j - FVC_i = 0 \\ -1\ FVC_j - FVC_i < 0 \end{cases} \tag{5}$$

$$Var(S) = \frac{n(n-1)(2n+5)}{18} \tag{6}$$

$$Z_c = \begin{cases} \frac{S-1}{\sqrt{var(S)}}, S > 0 \\ 0,\ \ S = 0 \\ \frac{S+1}{\sqrt{var(S)}}, S < 0 \end{cases} \tag{7}$$

In the above equations, $FVC_j$ and $FVC_i$ refer to the FVC values of the $j$ and $i$ years, respectively; $n$ represents 36 in this study; $sign$ is a symbolic function; and var( ) is the variance function of the random variable. The scope of Z is $(-\infty, +\infty)$.

Theil–Sen median trend analysis and the Mann–Kendall test were superimposed in ArcGIS10.4, and the superimposed results were divided into five classes, as shown in Table 2.

**Table 2.** Theil–Sen median trend analysis and the Mann–Kendall test level ($p < 0.05$) of trend change.

| β | Z | Trend Characteristics |
|---|---|---|
| β > 0 | Z > 1.96 | Significantly increased |
| | Z < 1.96 | Increased |
| β = 0 | Z = 0 | Stable and unchanged |
| β < 0 | Z > −1.96 | Decreased |
| | Z < −1.96 | Significantly decreased |

2.3.4. Multiple Regression Residual Analysis

Multiple linear regression and residual analysis were used to study the influences and relative contributions of climate variability and human activities to the changes in vegetation coverage [63] (this study assumed that FVC was only affected by the combined impact of two factors: human activities and climate variability). This method mainly involves the following three steps: (1) Selection of the growing season FVC as the dependent variable, with temperature and precipitation as the independent variables, establishes a binary linear regression model and permits the calculation of the parameters of the model. (2) Using the temperature and precipitation data, along with the parameters of the regression model, the predicted value of FVC ($FVC_{CC}$) was calculated (Equation (8)) to represent the impact of climate factors on the FVC. (3) The difference between the FVC observations ($FVC_{obs}$, FVC inverted from remote sensing images) and $FVC_{CC}$ was calculated and defined as the FVC residual ($FVC_{HA}$) [64,65]. The FVC residual ($FVC_{HA}$) represents the impact on the FVC in the context of human activities. The calculation formula is as follows:

$$FVC_{CC} = a \times Tmp + b \times Pre + c \tag{8}$$

$$FVC_{HA} = FVC_{obs} - FVC_{CC} \tag{9}$$

where $a$, $b$, and $c$ are regression model parameters, and $Tmp$ and $Pre$ denote the mean temperature and accumulated precipitation, respectively.

2.3.5. Correlation Analysis and Partial Correlation Analysis

Pearson's correlation coefficient was used to analyze the correlation between FVC and HA to characterize the response of vegetation coverage to HA. Pearson's correlation coefficient ($r$) was calculated as follows:

$$r_{xy} = \frac{\sum\limits_{i=1}^{N} \left[ \left( X_i - \overline{X} \right) \left( Y_i - \overline{Y} \right) \right]}{\sqrt{\sum\limits_{i=1}^{N} \left( X_i - \overline{X} \right)^2 \sum\limits_{i=1}^{N} \left( Y_i - \overline{Y} \right)^2}} \tag{10}$$

where $r_{xy}$ represents the degree of correlation between variables x and y.

When multiple factors are simultaneously correlated with FVC, partial correlation analysis [66] can eliminate the influence of other factors and separately analyze the correlation between a single factor and FVC. The formula for partial correlation analysis is as follows:

$$r_{xy.z_1 z_2 \cdots z_g} = \frac{r_{xy.z_1 z_2 \cdots z_{g-1}} - r_{xzg.z_1 z_2 \cdots z_{g-1}} r_{yzg.z_1 z_2 \cdots z_{g-1}}}{\sqrt{\left( 1 - r_{xzg.z_1 z_2 \cdots z_{g-1}}^2 \right) \left( 1 - r_{yzg.z_1 z_2 \cdots z_{g-1}}^2 \right)}} \tag{11}$$

where $r_{xy \cdot z1z2 \cdots zg}$ are the bias correlation coefficients of $z_1$, $z_2$, ..., $z_g$ of the partial correlation coefficients of the $x$ and $y$ variables of the control variables, respectively.

## 3. Results

### 3.1. Multivariate Residual Regression Model Rationality Assessment

The residual plot was used to verify the rationality of the hypothesis of the multiple regression residual model in this study. It could be seen that the residual followed a normal distribution ($p < 0.01$) (Figure 4a). Figure 4b indicates that the model assumptions in this study were reasonable. Moreover, the residuals were uncorrelated and random (Figure 4c).

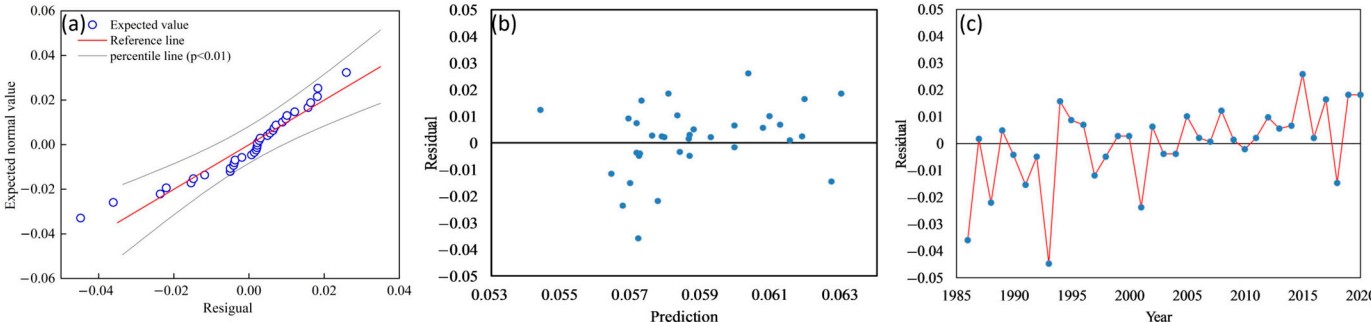

**Figure 4.** Residual plots. (**a**) Residual normal distribution. (**b**) Distribution of residuals and predicted values. (**c**) Time series distribution of residuals.

### 3.2. Spatiotemporal Variation Characteristics of FVC

The spatiotemporal change model of the FVC in the reserve was established using the dimidiate pixel model. The results showed that the annual average FVC of the reserve showed a significant upward ($p < 0.05$) trend from 1986 to 2021, with a linear trend of $1.7 \times 10^{-3} \text{ y}^{-1}$ (Figure 5a). The annual FVC mutation map of the reserve was constructed using the Mann–Kendall mutation test (Figure 5b). The results showed that this mutation occurred in 2009. This study used 2009 as the time node to segment the FVC time series of the reserve, as there was an effective mutation point in the UF and UB curves in 2009. The increasing trend of FVC in the reserve from 2010 to 2021 was slightly greater than that from 1986 to 2009. As shown in Figure 5c,d, the FVC-increased area accounted for about 28.39% of the total area from 1986 to 2009, mainly distributed in certain vegetation types, such as meadows and broad-leaved forests in Wuwei City. In contrast, the unchanged area accounted for about 27.70% of the total area, with higher distributions in the high-altitude areas of Zhangye City, where the land cover was mainly alpine meadows and perennial snow. The descending area accounted for approximately 43.91% of the total area and was mainly distributed in deserts and alpine sparse vegetation areas in high-altitude areas of Zhangye City. The percentage of FVC-decreased area was greater than that of the FVC-increased area during this time period. From 2010 to 2021, the increased area of FVC accounted for about 48.78%, mainly distributed in the grassland and meadow vegetation-type areas in some parts of Zhangye City, and the grassland vegetation area in Wuwei City. The unchanged FVC area accounted for approximately 16.53% of the total area, mainly distributed in the perennial snow-covered areas of Zhangye City, shrubs, meadows, and other vegetation types in Wuwei City. The FVC reduction area accounted for approximately 34.69% of the total area, mainly distributed in the grassland and meadow vegetation types in Zhangye City. In the past 12 years, the area of FVC increased more than the area decreased. Generally, the FVC of the reserve has significantly increased since 2009.

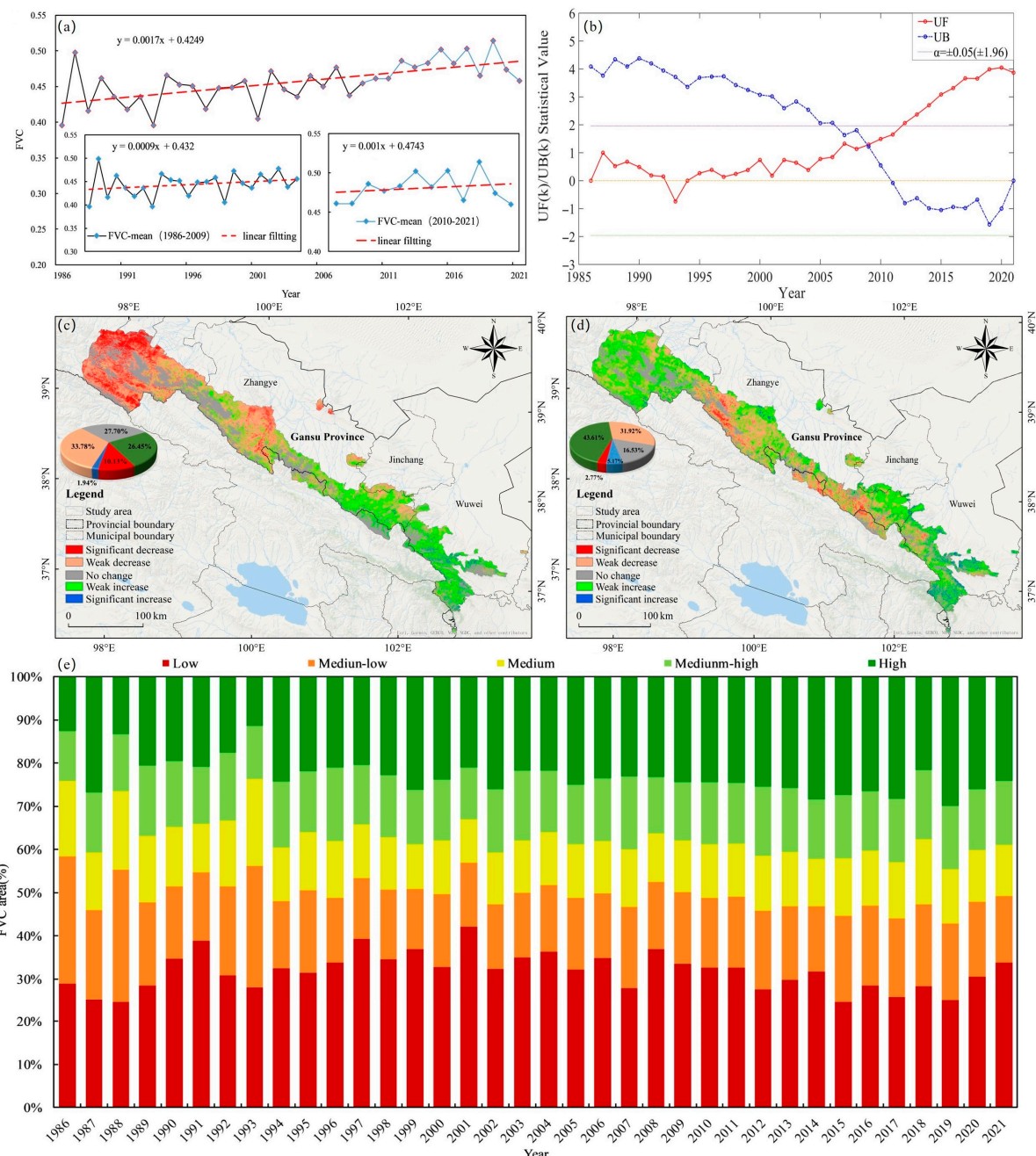

**Figure 5.** FVC interannual variation, Theil–Sen median trend analysis, and Mann–Kendall trend change and area proportion. (**a**) FVC trend change from 1986 to 2021, (**b**) the MK mutation test, (**c**) Theil–Sen median trend analysis, and Mann–Kendall trend change and area ratio from 1986 to 2009, (**d**) Theil–Sen median trend analysis and Mann–Kendall trend change and area ratio from 2010 to 2021, and (**e**) the area proportion from 1986 to 2021.

From the perspective of the city scale (Figure 6), the intensity of vegetation damage in each area of the reserve gradually decreased, and the FVC time series curve showed a fluctuating upward trend. However, some areas still deteriorated. The recovery rate of FVC in Jinchang City was the fastest, reaching $5.3 \times 10^{-3}$ y$^{-1}$, followed by the Wuwei section of the reserve, with a recovery rate of FVC at $2.7 \times 10^{-3}$ y$^{-1}$. The recovery rate of FVC in the Zhangye section of the reserve was the slowest ($1.3 \times 10^{-3}$ y$^{-1}$).

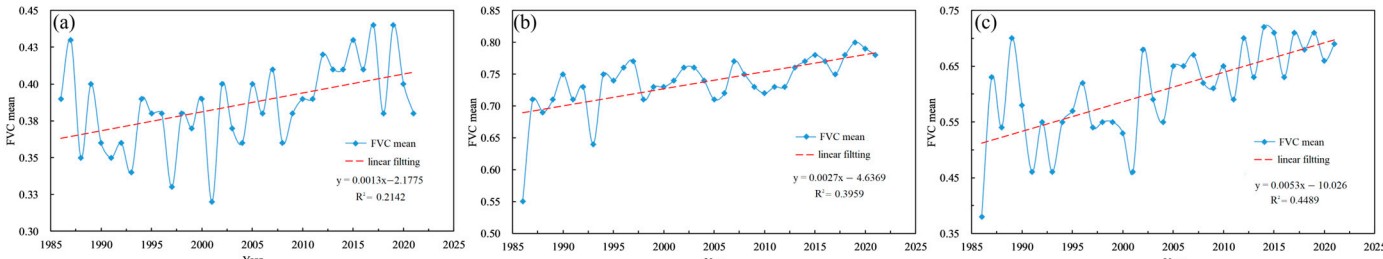

**Figure 6.** FVC changes at a city scale. (**a**) FVC trend change in the Zhangye section of the reserve from 1986 to 2021, (**b**) FVC trend change in the Wuwei section of the reserve from 1986 to 2021, and (**c**) FVC trend change in the Jinchang section of the reserve from 1986 to 2021.

### 3.3. Analysis of the Driving Factors of FVC

Spatial heterogeneity exists in the dynamic changes in the FVC in the reserve due to climate variability and human activities (Figure 7). The area where climate variability promoted vegetation improvement accounted for 43.34%, and was mainly distributed in the middle of the reserve. In contrast, the area of stable vegetation growth accounted for 53.11%, and was mainly distributed at both ends of the reserve. Vegetation-degraded areas in the reserve accounted for 3.55%. The area where human activities promoted vegetation improvement accounted for 50.76%, mainly distributed in the southeast part of the reserve. The area where vegetation grew stably accounted for 21.20%, and the area of vegetation degraded by human activities accounted for 28.04%, and was mainly distributed in the northwestern part of the reserve.

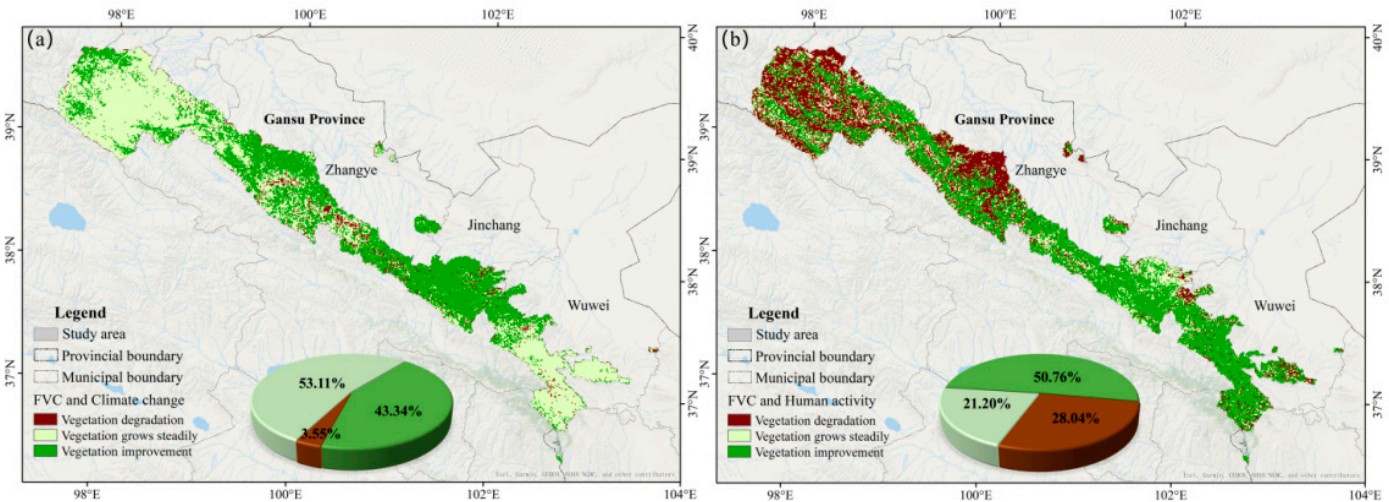

**Figure 7.** Spatial distribution of the impact of climate variability and human activities on FVC in the reserve from 1986 to 2020. (**a**) Impact of climate variability on the FVC. (**b**) Impact of human activities on the FVC.

### 3.3.1. Relative Contributions of Climate Variability and Human Activities to the FVC

As shown in Figure 8, the rate of climate variability contribution to the change in FVC in the reserve was 59% of that in the positive area. Areas with contribution rates greater than 80% accounted for 13%, and were mainly distributed in the border areas of Zhangye, Wuwei, and Jinchang. The contribution of climate variability to the change in FVC in the reserve accounted for approximately 41% of the negative area, which was mainly distributed in the high-altitude areas of Zhangye City.

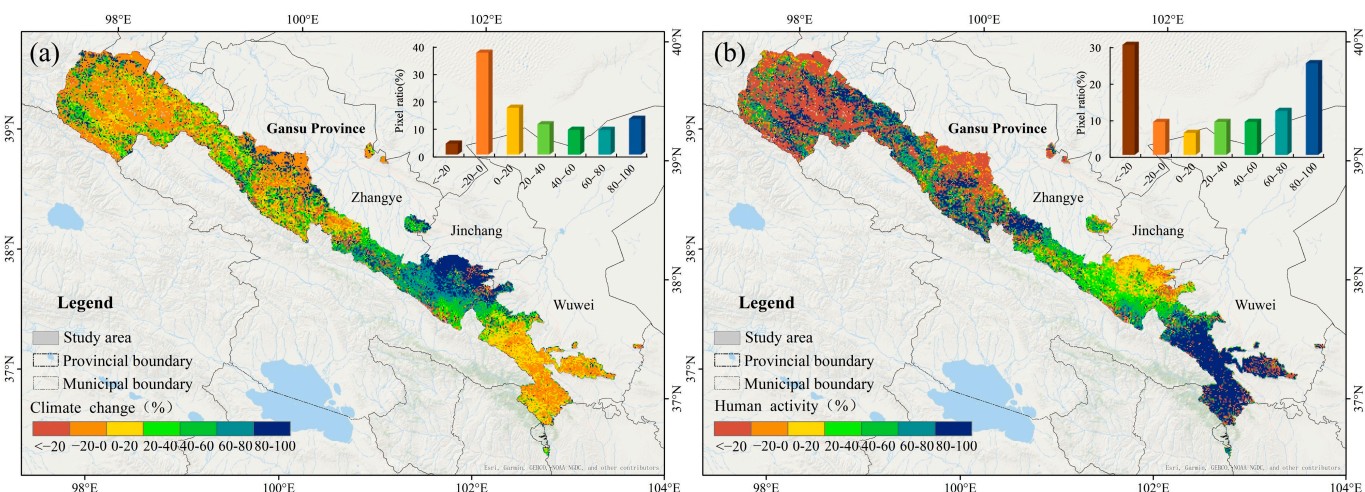

**Figure 8.** Spatial distributions of the relative contribution rates of climate variability and human activities to FVC in the reserve. (**a**) Relative contribution of climate variability to the FVC. (**b**) Relative contribution of human activities to the FVC.

The contribution of human activities to the change in FVC in the reserve was 61% of that in the positive area. The area with a contribution rate greater than 80% was 25% and was mainly distributed in the low-altitude areas of Wuwei City in the reserve. The contribution of human activities to the change in FVC in the reserve accounted for approximately 39% of the negative area and was mainly distributed in the high-altitude areas of the reserve.

### 3.3.2. Spatiotemporal Evolution Characteristics of FVC under the Influence of Climate Variability

Long-term changes in temperature and precipitation in the reserve from 1986 to 2020 were analyzed. The results showed that since 1986, the reserve has experienced a noticeable trend of warming and humidification (Figure 9), providing good conditions for vegetative growth and recovery. From 1986 to 2020, the temperature in the reserve showed a significant upward trend, with an average temperature of –0.89 °C, rising at a rate of $4.5 \times 10^{-2}$ y$^{-1}$. After 1997, the climate warming rate significantly increased ($p < 0.05$). From 1986 to 2020, the precipitation in the reserve showed a significant increasing trend, with an average precipitation of 177.86 mm, increasing at a rate of $6.82 \times 10^{-1}$ y$^{-1}$.

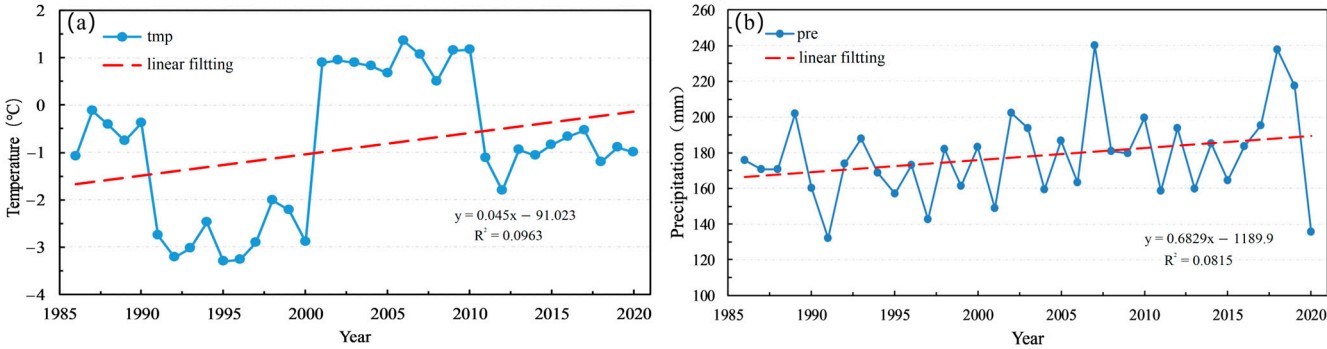

**Figure 9.** Temperature and precipitation changes from 1986 to 2020. (**a**) Temperature changes in the reserve from 1986 to 2020. (**b**) Precipitation changes in the reserve from 1986 to 2020. (tmp represents temperature, and pre represents precipitation).

Figure 10 shows that FVC in the reserve was positively correlated with temperature and precipitation. The partial correlation coefficient between the FVC and precipitation

was 0.155. The positive correlation between the FVC and precipitation was about 75.11%, mainly distributed in the northwestern end of the reserve. The mean value of the bias relationship between the FVC and temperature was lower than that for precipitation (0.027). The proportion of FVC positively correlated with temperature was 51.08% and was mainly distributed in the southeastern end of the reserve.

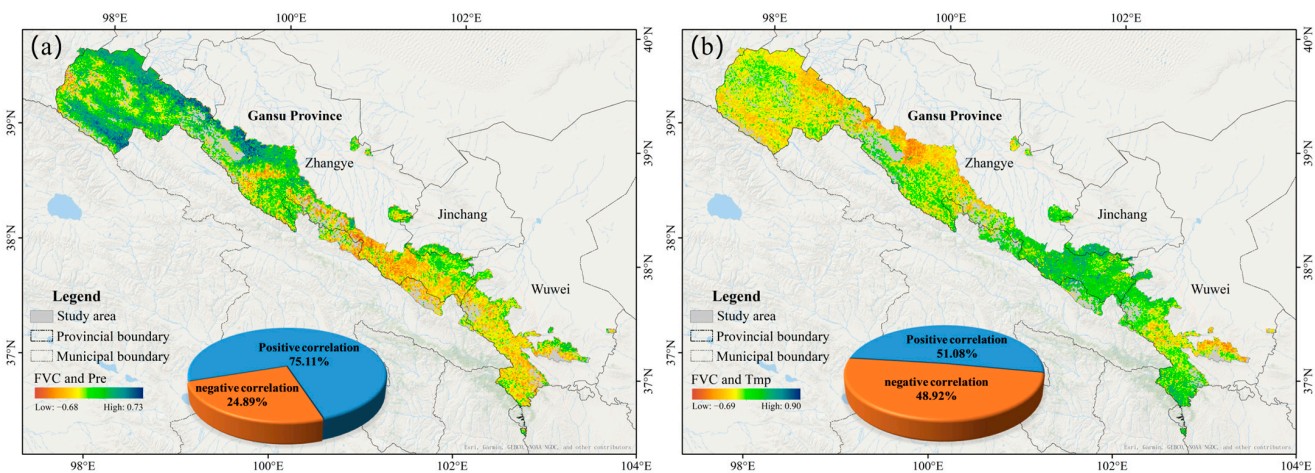

**Figure 10.** Partial correlation between the FVC and climate. (**a**) FVC and precipitation partial correlation. (**b**) FVC and temperature partial correlation. tmp represents temperature, and pre represents precipitation.

### 3.3.3. Spatiotemporal Evolution of FVC under the Influence of Human Activities

The human footprint alters natural ecological processes. The landscape changes exert pressure on the ecological environment [47]. The global annual human activity footprint data show that the overall human activities in the reserve showed a downward trend from 2000 to 2018, with a decline rate of $1.9 \times 10^{-3}$ $y^{-1}$ (Figure 11b). Pearson's correlation coefficient was used to explore the influence of human footprints on the FVC of the reserve (Figure 11c). The area where FVC was positively correlated with human activity footprints was approximately 41.78%, with a significant positive correlation of 5.10%. However, the negative correlation area was approximately 58.22%, with a significant negative correlation area of 19.88%. The negative correlation areas were mainly distributed in the northwestern part of the reserve.

### 3.4. Consistency Test

FVC products of the National Qinghai–Tibet Plateau Scientific Data Center, with a monthly 30 m from 2019 to 2021, were used as validation data [67]. The effective pixel values of it, and the FVC data of the reserve, are shown in Figure 12 (N = 30,000). The results indicated that the coefficient of determination, $R^2$, was greater than 0.88. The FVC retrieved in this study was similar to the FVC product results, which also proved the liability of the results of this study.

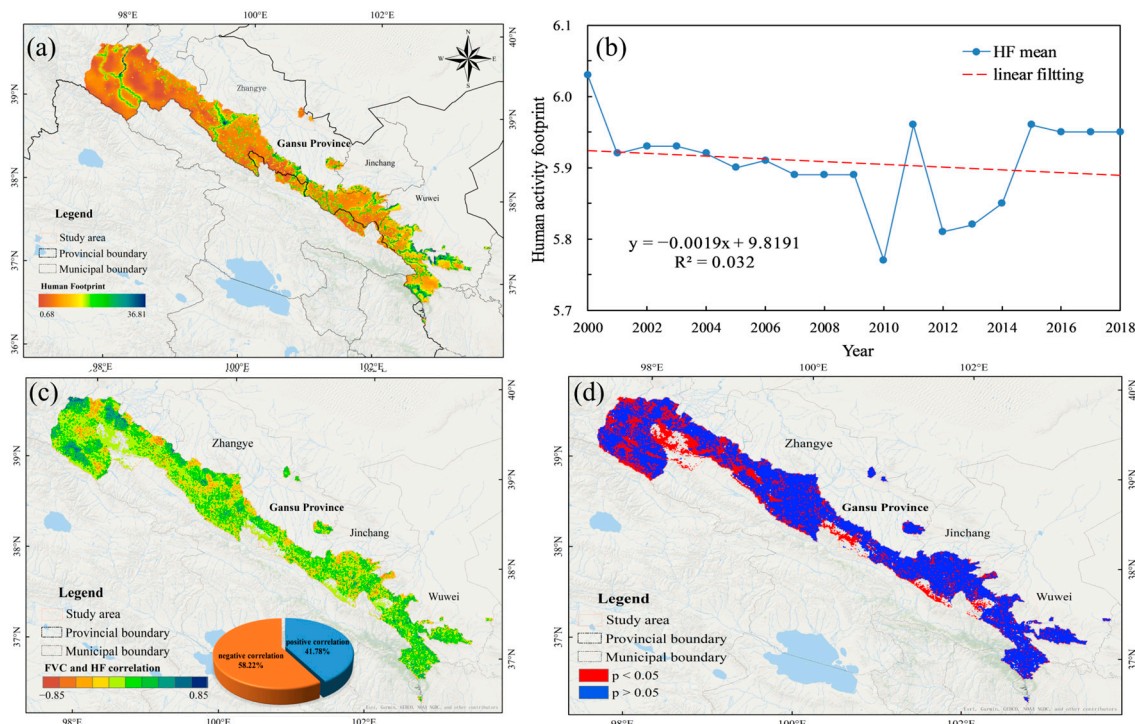

**Figure 11.** Spatiotemporal changes of the human activity footprint and its correlation with the FVC. (**a**) Spatial distribution of the human activity footprint. (**b**) Changes in the footprint of human activities trends from 2000 to 2018. (**c**) Correlation and area ratio between the human activity footprint and FVC. (**d**) The human activity footprint and FVC significance (HF refers to the human activity footprint).

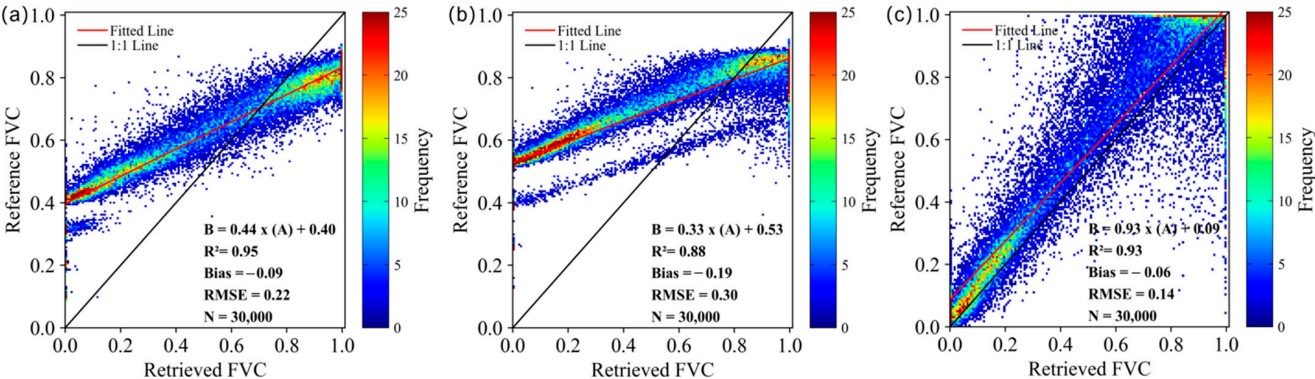

**Figure 12.** Consistency test ((**a**) 2019, (**b**) 2020, and (**c**) 2021).

## 4. Discussion

### 4.1. Spatiotemporal Variation Characteristics of FVC

The overall FVC of the reserve showed an upward trend, and the growth rate of the FVC from 1986 to 2021 was $1.7 \times 10^{-3}$ $y^{-1}$. Using the Mann–Kendall mutation test, it was found that 2009 was the mutation year, and is consistent with existing research results [49,55,68]. Although the vegetation condition improved, the rate of FVC growth from 1986 to 2009 was $0.9 \times 10^{-3}$ $y^{-1}$, as the rate was slow. The rate of FVC growth from 2010 to 2021 was $1.0 \times 10^{-3}$ $y^{-1}$, which is faster than that between 1986 and 2009. In 2003, the Chinese government implemented a national conservation policy to support the ecological project of the degrazing plan in Northern China. In the same year, the local government of the Qilian Mountains launched an ecological protection project to transfer

52,000 herdsmen to the valley area within 3–5 years, which had a great impact on improving vegetation [49].

From 1986 to 2021, the FVC in the reserve showed a spatial distribution pattern of "low vegetation in the northwest and high vegetation in the southeast,", mainly because the southeastern part of the reserve has a lower latitude, higher temperature, and more precipitation, which were conducive to the development of vegetation. From 1986 to 2009, the proportion of FVC increased by 28.39%, and the area proportion was relatively low, which was related to ecological damage phenomena, such as overgrazing, illegal construction of hydropower stations, illegal construction of mining and prospecting, and the disorderly operation of tourist facilities. From 2010 to 2021, the FVC growth area accounted for 48.78%, and the area increased. The rising rate of forestland in the reserve was $0.34 \, \text{y}^{-1}$, and the rising rate of grassland was $0.18 \, \text{y}^{-1}$. The mine environment was fully restored through implementing certain measures, such as afforestation, fencing, soil leveling, and reinforced berms [69]. A good policy atmosphere, and effective policy implementation, restored the vegetation and ecological environment in the reserve.

From a city-scale perspective, compared with the other two cities, the average value of FVC in Zhangye City was lower. The *Statistical Yearbook of Gansu Province*, along with previous studies [69], revealed that many unauthorized hydroelectric power stations, tourist attractions, and mining prospecting projects exist in the Zhangye section of the reserve, and relatively high-intensity human sabotage activities have caused severe vegetative damage. Simultaneously, the vegetation types in the Zhangye section of the reserve were mainly shrubs, snow-covered glaciers, and other vegetation types, and the vegetation recovery period was relatively long.

### 4.2. Analysis of the Driving Factors of FVC

#### 4.2.1. Relative Contributions of Climate Variability and Human Activities to the FVC

The contribution rate of climate variability to the change in FVC in the reserve was positive, accounting for 59%. Recent studies have shown that the reserve is located in arid and semi-arid regions, where climate transitions occurred from warm and dry to warm and humid [70]. As most arid and semi-arid regions have insufficient precipitation to meet the needs of vegetative growth, this warming and humidification pattern suggests that climate variability may provide a benefit to vegetative growth in the reserve [71]. The contribution rate of human activities to the change in FVC in the reserve was positive, accounting for 61% of the area. The reserve was established in 1986, and the Grain for Green Program policy was implemented in 2003. With the concept that the lucid water and lush mountains are invaluable assets and the environmental protection policy of mountains, rivers, forests, farmlands, lakes, and grasslands were implemented successively, laying an important foundation for restoring vegetation in the reserve.

#### 4.2.2. Evolution Characteristics of FVC under the Influence of Climate Variability

From 1986 to 2020, the temperature increase rate in the reserve was $4.5 \times 10^{-2} \, \text{y}^{-1}$, and the precipitation increase rate was $6.82 \times 10^{-1} \, \text{y}^{-1}$. The warming and humidification phenomena in the reserve were evident. Partial correlation analysis showed that among the climate variability factors, precipitation was the dominant factor affecting vegetation. At the northwestern part of the reserve, precipitation was positively correlated with FVC [55], whereas temperature was negatively correlated, as an increase in temperature may increase evaporation and reduce soil moisture, thereby limiting vegetative growth [72]. Precipitation was negatively correlated with FVC at the southeastern part of the reserve, whereas temperature was positively correlated. Excessive precipitation aggravates soil erosion and reduces soil organic matter content. An increase in temperature increases evapotranspiration, reduces the water use efficiency of plants [73,74], and promotes vegetative growth.

### 4.2.3. Human Activities affect the Spatiotemporal Evolution of FVC

The decline rate of the human activity footprint is $1.9 \times 10^{-3}$ y$^{-1}$, which is consistent with existing research data [75], and the intensity of human activities reached its maximum in 2016. Pearson's correlation coefficient showed that the area of positive correlation between FVC and human activity footprints accounted for approximately 42% and was mainly distributed in the low-altitude areas of the Wuwei section, where the grazing pressure is low, and the vegetation types are mainly cultivated vegetation. Good geographical conditions and low livestock pressures are advantageous for vegetative growth. The area of negative correlation between FVC and human activities accounted for approximately 58%, and was mainly distributed in areas, such as Zhangye City, within the reserve where the altitude is high, and where there are many types of alpine vegetation, shrubs, and meadow vegetation. The pressure on livestock is high, and the disorderly mining of mines and illegal construction of hydropower stations have destroyed the ecological environment and inhibited the growth and restoration of vegetation to a large extent, which is consistent with the fact that human activities inhibit vegetative growth, mainly in areas such as Zhangye City, in the reserve.

### 4.3. Limitations of the Current Study

The drivers of vegetative growth and change are critical to the sustainable management and development of reserve ecology. Although the SR data in Landsat Collection 2 Tiers 1 have been calibrated between sensors, the atmospheric and surface noise signals, along with the choice of models, may lead to uncertainty in the identification of vegetation coverage [76,77]. In this study, the normalized difference vegetation index (NDVI) was used to invert the vegetation coverage. However, since the NDVI has a non-linear dependence on leaf overlap, the vegetation coverage will be represented in a non-linear manner for sparse-leaf and dense-leaf vegetation types. In future research, we will try to improve or choose better indicators for vegetation coverage extraction.

The multiple regression residual analysis method was widely used to study the effects of human activities on vegetation changes. However, the method itself also had certain drawbacks [44]. At present, it was not clear how to reasonably select climate factors. Previous studies have shown that vegetation changes in the Qinghai–Tibet Plateau was mainly influenced by temperature and precipitation [21,22]. However, climate variability also includes climate factors, such as humidity, wind speed, sunlight duration, solar radiation, and so on [78]. Therefore, more climate factors should be used to analyze changes in vegetation changes in the study area for future research.

## 5. Conclusions

In this study, the FVC was used as an index to monitor the dynamic FVC changes in the reserve. The effects of climate variability and human activities on vegetation dynamics were assessed using partial correlation and multiple regression residual analyses. The results show that:

(1) From 1986 to 2021, FVC in the reserve recovered in stages. A sudden change happened in 2009, and the increase rate of FVC in 2010–2021 was greater than that in 1986–2009. From 1986 to 2021, high vegetation coverage of the FVC was mainly distributed in the southeastern part of the reserve, and low vegetation coverage was mainly distributed in the northwestern part of the reserve. Due to the high intensity of vegetation damage in the Zhangye section of the reserve, the vegetation recovery period was relatively long.

(2) Climate variability and human activities have obvious spatial heterogeneity on FVC changes. Climate variability contributed 49% to the increase in FVC in the reserve, and human activities contributed 51% to the increase in FVC in the reserve, dominating the growth of FVC in most areas. Multiple regression residual analysis can quantify the impact of climate change and human activities on vegetation, but how to reasonably select climate elements is important.

(3) The warming and humidification phenomena in the reserve are evident. In climate variability, precipitation is the dominant factor affecting vegetation change, followed by temperature. The areas positively correlated with precipitation were mainly distributed in the high-altitude areas of Zhangye City in the reserve. The areas positively correlated with temperature were mainly distributed at the junction of Zhangye and Wuwei. Overall, affected by human activities and climate variability, the FVC in the reserve has increased annually, and the ecological environment has tended to improve.

**Author Contributions:** Conceptualization, X.W. (Xiaoxian Wang) and X.Z. (Xiuxia Zhang); methodology, X.W. (Xiaoxian Wang) and X.Z. (Xiuxia Zhang); software, X.W. (Xiaoxian Wang) and W.L.; validation, X.W. (Xiaoxian Wang) and X.C.; formal analysis, X.W. (Xiaoxian Wang) and X.C.; investigation, Z.Z.; resources, W.L., J.H. and X.L.; data curation, X.W. (Xiaoxian Wang); writing—original draft preparation, X.W. (Xiaoxian Wang); writing—review and editing, X.Z. (Xiuxia Zhang), Y.L. and X.W. (Xiaodong Wu); visualization, X.W. (Xiaoxian Wang), L.D. and X.Z. (Xilai Zhang); supervision, X.Z. (Xiuxia Zhang) and Q.L.; project administration, W.L.; funding acquisition, W.L. All authors have read and agreed to the published version of the manuscript.

**Funding:** This work was supported by the Natural Science Foundation of Gansu Province, grant number 22JR5RA247; the Natural Science Foundation of Gansu Province, grant number 21JR7RA242; the Educational Science and Technology Innovation Project of Gansu Province, grant number 2022QB—052; the Educational Science and Technology Innovation Project of Gansu Province, grant number 2022QB—046.

**Data Availability Statement:** The data are unavailable due to privacy and ethical restrictions.

**Conflicts of Interest:** The authors declare no conflict of interest.

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
