# Peer review of "Quantitative Analysis of Climate Variability and Human Activities on Vegetation Variations in the Qilian Mountain National Nature Reserve from 1986 to 2021"

_forests, doi:10.3390/f14102042_

Round 1

Reviewer 1 Report

I can confidently affirm that the manuscript has been skillfully crafted by the authors. I strongly recommend its publication, contingent upon the rectification of the noted deficiencies.

In the passage spanning lines 90 to 93, you introduce the opportunities afforded by GEE, including the capability to work with high-resolution images. Nevertheless, your paper exclusively employs Landsat imagery. It is pertinent to inquire why alternative sources, such as Sentinel imagery, offering superior resolution data, were not explored. I recommend revising your paper to address this query preemptively.

Line 188: Normalized Difference Vegetation Index (NDVI).

The subject matter addressed in the conclusion section of the manuscript essentially recapitulates information disseminated in the results and discussion sections, rendering the numerical values redundant. Hence, I recommend excluding such duplicative content from the Conclusion section. What is anticipated from the authors in this section is a comprehensive appraisal of the conducted study. It is crucial to elucidate the extent of its contribution towards ecological restoration, including its merits and demerits. Additionally, an endeavor should be made to differentiate your work from existing literature, enabling readers to discern its distinctiveness. Furthermore, the conclusion section should delineate the deficiencies and lacunae in the results, accompanied by proposals for rectification.

Line 16-17: In order to quantify the impact of climate change and human activities on the vegetation dynamics in the Qilian Mountains region.

Line 100: Based on...

On numerous occasions within the text, the subsequent sentence commences without the insertion of a space following the concluding punctuation mark. I kindly request a thorough review of the entire manuscript for this issue.

Author Response

Response to Reviewer 1 Comments

Point 1:

In the passage spanning lines 90 to 93, you introduce the opportunities afforded by GEE, including the capability to work with high-resolution images. Nevertheless, your paper exclusively employs Landsat imagery. It is pertinent to inquire why alternative sources, such as Sentinel imagery, offering superior resolution data, were not explored. I recommend revising your paper to address this query preemptively.

Respond 1:

Thanks for your suggestion. In October 1986, the Qilian Mountain National Nature Reserve was approved and established. In 2000, the reserve was designated as a national natural forest protection project area. In 2008, the reserve was designated as a water conservation ecological functional area. In order to more comprehensively monitor the changes in vegetation dynamics before and after the establishment of the protected area, it is also considered that the time series of Landsat remote sensing images is relatively long. Therefore, this paper conducted relevant research using Landsat data based on the GEE platform.

Point 2: 

Line 188: Normalized Difference Vegetation Index (NDVI).

The subject matter addressed in the conclusion section of the manuscript essentially recapitulates information disseminated in the results and discussion sections, rendering the numerical values redundant. Hence, I recommend excluding such duplicative content from the Conclusion section. What is anticipated from the authors in this section is a comprehensive appraisal of the conducted study. It is crucial to elucidate the extent of its contribution towards ecological restoration, including its merits and demerits. Additionally, an endeavor should be made to differentiate your work from existing literature, enabling readers to discern its distinctiveness. Furthermore, the conclusion section should delineate the deficiencies and lacunae in the results, accompanied by proposals for rectification.

Respond 2:

Thanks for your suggestion. We have made relevant changes to the conclusion part of the paper. At the same time, we also added the shortcomings and prospects of this study in Section 4.3 of the revised manuscript discussion.

The multiple regression residual analysis method was widely used to study the effects of human activities on vegetation changes. However, the method itself also had certain drawbacks. At present, it was not clear how to reasonably select climate factors. Previous studies had shown that vegetation changes in the Qinghai Tibet Plateau was mainly influenced by temperature and precipitation. However, climate variability also includes climate factors such as humidity, wind speed, sunlight duration, solar radiation and so on. Therefore, more climate factors would be used to analyze changes in vegetation changes in the study area for future research.

See lines 488-495 of the manuscript for details.

(1)During 1986–2021, FVC in the reserve recovered in stages. A sudden happened in 2009, and the increase rate of FVC in 2010–2021 was greater than that in 1986–2009. During 1986–2021, high vegetation coverage of the FVC was mainly distributed in the southeastern part of the reserve, and low vegetation coverage was mainly distributed in the northwest part of the reserve. Due to the high intensity of vegetation damage in the Zhangye section of the reserve, the vegetation recovery period was relatively long.

(2) Climate variability and human activities have obvious spatial heterogeneity on FVC changes. climate variability contributed 49% to the increase of FVC in the reserve, and human activities contributed 51% to the increase in FVC in the reserve, dominating the growth of FVC in most areas. Multiple regression residual analysis can quantify the impact of climate change and human activities on vegetation, but how to reasonably select climate elements is important.

(3) The warming and humidification phenomenona in the reserve are evident. In climate variability, precipitation is the dominant factor affecting vegetation change, followed by temperature. The areas positively correlated with precipitation were mainly distributed in the high-altitude areas of Zhangye City in the reserve. The areas positively correlated with temperature were mainly distributed at the junction of Zhangye and Wuwei. Overall, affected by human activities and climate variability, the FVC in the reserve has increased annually, and the ecological environment has tended to improve.

See lines 501-512 of the manuscript for details.

Point 3: 

Line 16-17: In order to quantify the impact of climate change and human activities on the vegetation dynamics in the Qilian Mountains region.

Line 100: Based on...

On numerous occasions within the text, the subsequent sentence commences without the insertion of a space following the concluding punctuation mark. I kindly request a thorough review of the entire manuscript for this issue.

Respond 3:

Thanks for your suggestion. Sorry, This is a mistake we made while writing the paper. We have checked and revised the entire manuscript, and the revised parts have been marked in red font.

Reviewer 2 Report

Dear Authors,

I have reviewed the manuscript entitled "Quantitative Analysis of Climate Change and Human Activities on FVC Variations in the Qilian Mountain National Nature Reserve During 1986-2021", submitted for consideration for publication in the Forests (MDPI) Journal. Based on my review, I believe that the manuscript has potential, and conveys interesting findings. Moreover, the topic is of concern and the scope matched the targetted journal.

I would like to remain highly positive on my review. Therefore, I have provided a wealth of comments to outline the potential limitations, aspects to clarify/improve throughout the entire manuscript. Please provide a point-by-point response to each of these comments. Also, please go through the entire manuscript for proofreading, as there are many style and grammar issues. In many places, there are unfinished sentences that could be further cleared up and improved.

My definitive opinion is "Major Review".

Please go through the entire manuscript for proofreading, as there are many style and grammar issues. In many places, there are unfinished sentences that could be further cleared up and improved.

Author Response

Response to Reviewer 2 Comments

Point 1:

 Throughout the paper, I believe that there is a misuse of the term “climate change”. Change is adapted for cases when different periods are ultimately compared, or when at least, a distant future period is compared to a current period. In this case, authors are evaluating over the recent period how the climate patterns/fluctuations are affecting FVC variations as well. I believe that authors should refer to “climate variability” here, instead of “climate change”

Response 1:

Thanks for your suggestion. We have changed all “climate change” in the article to “climate variability”.

Point 2:

Response 2:

Thanks for your suggestion. Fractional Vegetation Coverage(FVC) is an important indicator for monitoring the growth status of surface vegetation and reflecting changes in ecological environment quality. Based on your valuable opinions, we have replaced "FVC" with "Vegetation" in the title of the revised manuscript.

Point 3:

Respond 3:

Thanks for your suggestion. We rechecked sentences for completeness and grammar, and made revisions. Thank you again for your valuable advice and have a nice life.

The time series of fractional vegetation coverage (FVC) from 1986 to 2021 were used to quantify the impact of climate variability and human activities on vegetation variations in the Qilian Mountain National Nature Reserve(QMNNR), using 3147 land satellite images based on the Google Earth Engine cloud platform.

See lines 16-19 of the revised manuscript for details.

We selected Landsat 5/7/8 SR data based on the Landsat Collection Tier 1 dataset available online on the GEE platform.

See lines 152-153 of the revised manuscript for details.

Point 4:

 How could this be translated into practical recommendations?

Respond 4:

Thank you for your suggestion. Our study found that human activities contribute more to vegetation than climate variability. This shows that a series of ecological projects are beneficial to the restoration of vegetation in the reserve. Therefore, in future ecological protection work, we will be able to recommend that relevant government departments develop more man-made ecological protection policies and measures. At the same time, this study improves people's understanding of the response of vegetation dynamic changes in the Qilian Mountains National Nature Reserve to climate and human activities.

See lines 32-33 of the revised manuscript for details.

Point 5:

Respond 5:

Thanks for your suggestion. By reading this article, we better understand the importance of vegetation in promoting surface water circulation and the impact of surface water on vegetation growth. We also cited this high-quality literature in our revised manuscript.

See the revised manuscript reference [2] for details.

[2] Kafando M B, Koïta M, Le Coz M, et al. Use of multidisciplinary approaches for groundwater recharge mechanism characterization in basement aquifers: case of sanon experimental catchment in Burkina Faso[J]. Water, 2021, 13(22): 3216.

Point 6:

Respond 6:

Thanks for your suggestion. We re-summarized relevant references and revised the manuscript.

Recent research shows that "greening the planet" is prominent in China and India due to the impact of climate variability and human activities.

See lines 43-45 of the manuscript for details.

Point 7:

Respond 7:

Thanks for your suggestion. We replaced "change" with "variability" in the manuscript.

Point 8:

That claim is not really true. A single and quick Google Scholar search will result in many applications, even in China. Authors could hedge a little, mention some of these previous applications but further specify that the study area in which they are focusing have not been studied yet in this regard.(Line 89)

Respond 8:

Thanks for your suggestion. We revised the manuscript again.

The NDVI data obtained by using landsat has a longer time series and better spatial resolution and can be used to monitor vegetation changes in the QMNNR.

See lines 88-90 of the revised manuscript for details.

Point 9:

Respond 9:

Thanks for your suggestion. We have modified the revised manuscript based on your valuable comments.

In recent years, various methods have been used to analyze the attribution of climate variability and human activities to vegetation.

See lines 93-95 of the revised manuscript for details.

Point 10:

Respond 10:

Thanks for your suggestion. This is a small mistake that we made when writing our paper. We made the revisions in the revised manuscript.

Statistical methods have high requirements on the completeness and accuracy of historical statistical data and a single method cannot fully clarify the causes of vegetation changes. Combining statistical methods and partial correlation analysis methods can clearly quantify the factors affecting vegetation changes.

See lines 96-100 of the revised manuscript for details.

Point 11:

Respond 11:

Thanks for your suggestion. We filtered remote sensing images with cloud cover less than 20% based on GEE. We also made modifications in the revised manuscript.

See line 154 of the revised manuscript for details.

Point 12:

Respond 12:

Thanks for your suggestion. We revised the revised manuscript in detail for problems that arose.

Table 1. Sources of data used in this study

Datasets

types

Image usability analysis

Spatial resolution/m

Time resolution/year

Data source

Image data

Landsat 5 SR

raster

1326 scenes

30

1986-2011

United States Geological Survey https://www.usgs.gov/

Landsat 7 SR

raster

1167 scenes

30

1999-2021

United States Geological Survey

https://www.usgs.gov/

Landsat 8 SR

raster

654 scenes

30

2013-2021

United States Geological Survey

https://www.usgs.gov/

Basic data

Landsat

Path Row

(WRS–2)

vector

/

/

1983-now

Geodata Platform, School of Urban and Environmental Studies, Peking University

http://geodata.pku.edu.cn

Product data

raster

/

30

2019-2021

National Qinghai-Tibet Plateau Scientific Data Centre

https://data.tpdc.ac.cn/zh-hans/

Temperature and precipitation data

raster

/

1000

1986-2020(monthly)

Climatic Research Unit gridded Time Series

https://crudata.uea.ac.uk/cru/data/hrg/

Human footprint dataset

raster

/

1000

2000-2018

[47]

See table 1 of the revised manuscript for details.

Point 13:

Respond 13:

Thanks for your suggestion. We have reworked the abbreviations in the revised manuscript.

Construction of a dimidiate pixel model and spatial and temporal trend analysis using Theil-Sen analysis and Mann-Kendall test.

See lines 175-176 of the revised manuscript for details.

Point 14:

Respond 14:

Thanks for your suggestion. Based on the opinions of two external review experts, we changed the title of Figure 3 to “Flowchart showing the four main steps of this study”.

Point 15:

Respond 15:

Thanks for your suggestion. We have added a description of significance levels in the revised manuscript(Line 236). The criteria for trend analysis have also been modified based on your comments.

Table 2. Theil–Sen Median Trend Analysis and Mann–Kendall Test level (p < 0.05) of trend change

β

Z

Trend characteristics

β > 0

Z > 1.96

Significantly increase

Z < 1.96

increase

β = 0

Z = 0

Stable and unchanged

β < 0

Z > -1.96

decrease

Z < -1.96

Significantly decrease

See revised manuscript Table 2 for details.

Point 16:

Respond 16:

Sun L, Li H, Wang J, et al. Impacts of Climate Change and Human Activities on NDVI in the Qinghai-Tibet Plateau[J]. Remote Sensing, 2023, 15(3): 587.

Ma M, Wang Q, Liu R, et al. Effects of climate change and human activities on vegetation coverage change in northern China considering extreme climate and time-lag and-accumulation effects[J]. Science of The Total Environment, 2023, 860: 160527.

Point 17:

Respond 17:

The residual plot was used to verify the rationality of the hypothesis of the multiple regression residual model in this study. It could be seen that the residual followed a normal distribution (p < 0.01) (Figure 4a). Figure 4b indicated that the model assumptions in this study were reasonable. And the residuals were uncorrelated and random (Figure 4c).

Reviewer 3 Report

This study analyzes the spatio-temporal impacts of the climate change and land use activities on fraction of vegetation cover derived from the 1986-2021 Landsat time series over the Qilian Mountain National Nature Reserve in China. I found the paper interesting and enjoyed reading the paper. Overall, the paper is well written, and the quality of the figures is excellent. Results from statistical analyzes are convincing.  I only have the following minor comments:

1. Currently, the primary objective of the paper is not clear. According to the title, it is quantitative analysis of climate change and human activities on FVC variations in the Qilian Mountain National Nature Reserve; in the Introduction section (L130-132), it is temperature, precipitation, and human activity data analyses to quantify the impact of human activities and climate change on vegetation. Please, try to be more consistent with the objective of the study.

2. In the Data Source section (L153), please inform if the Landsat collection used in this study was Tier 1 or Tier 2.

3. The title of Figure 3 can be improved. My suggestion: Flowchart showing the four main steps of this study. All abbreviations included in the figure should be stated in the title. Same for other figures.

4. Proper credit should be given to the authors who proposed the FCV estimation from NDVI, as described in the manuscript by Eq. 2: Qi et al. Spatial and temporal dynamics of vegetation in the San Pedro River basin area. Agricultural and Forest Meteorology, v. 105, p. 55−68, 2000. The advantages of using FCV rather than using NDVI itself should be stated in the manuscript.

5. In the Results section, the authors go directly to the time series change analyses. In the first part of this section, it may be a good idea to show a figure with the average percentage of FVC at the Landsat pixel level for the National Reserve during the considered period. Thus, the readers can have better idea about the fraction of vegetation cover of the study area.

 6. I would not expect two drastic decreases in the temporal behavior of temperature over the period, as shown in Figure 9. I expect much more similar behavior as shown by precipitation. I suggest checking this data with some other data sets available in the internet or, if known, provide some explanation about this temporal signature.

English writting is fine, minor editing is demanded.

Author Response

Response to Reviewer 3 Comments

Point 1:

 Currently, the primary objective of the paper is not clear. According to the title, it is quantitative analysis of climate change and human activities on FVC variations in the Qilian Mountain National Nature Reserve; in the Introduction section (L130-132), it is temperature, precipitation, and human activity data analyses to quantify the impact of human activities and climate change on vegetation. Please, try to be more consistent with the objective of the study.

Response 1:

Thanks for your suggestion. Taking into account the opinions of another external reviewer, we revised the title of the manuscript to "Quantitative Analysis of Climate Variability and Human Activities on Vegetation Variations in the Qilian Mountain National Nature Reserve During 1986-2021". Therefore, the main research goal of this paper is to quantify the changes in vegetation in the Qilian Mountains National Nature Reserve caused by climate change and human activities. Because FVC is better able to measure surface vegetation conditions and ecological environment changes. We use FVC as an indicator to reflect vegetation changes in the reserve.

The Qilian Mountains National Nature Reserve is located in the northeastern part of the Qinghai-Tibet Plateau. Existing studies have shown (Chen et al., 2020; Huang et al., 2016) that vegetation in the Tibetan Plateau is mainly affected by temperature and precipitation. Therefore, we discuss the impact of two main factors in climate change (temperature and precipitation) on vegetation in the study area in the introduction of the manuscript (Lines 56-59)..

Point 2:

In the Data Source section (L153), please inform if the Landsat collection used in this study was Tier 1 or Tier 2.

Response 2:

Thanks for your suggestion. The Landsat collection used in this study was Tier 1.

We selected Landsat 5/7/8 SR data based on the Landsat Collection Tier 1 dataset available online on the GEE platform.

See Line 152-153 in the revised manuscript for details.

Point 3:

The title of Figure 3 can be improved. My suggestion: Flowchart showing the four main steps of this study. All abbreviations included in the figure should be stated in the title. Same for other figures.

Respond 3:

Thanks for your suggestion. We have modified the article based on your comments and added an explanation of the abbreviation in the figure title.

Point 4:

 Proper credit should be given to the authors who proposed the FVC estimation from NDVI, as described in the manuscript by Eq. 2: Qi et al. Spatial and temporal dynamics of vegetation in the San Pedro River basin area. Agricultural and Forest Meteorology, v. 105, p. 55−68, 2000. The advantages of using FVC rather than using NDVI itself should be stated in the manuscript.

Respond 4:

Thank you for your suggestion. We read that literature in detail. We made changes in the revised manuscript and cited this article.

FVC is better able to measure surface vegetation conditions and ecological environment changes[58].

Please see lines 202-203 of the manuscript and the literature [58].

Thank you again for your authoritative advice and have a great life.

[58] Qi J, Marsett R C, Moran M S, et al. Spatial and temporal dynamics of vegetation in the San Pedro River basin area[J]. Agricultural and forest meteorology, 2000, 105(1-3): 55-68.

Point 5:

In the Results section, the authors go directly to the time series change analyses. In the first part of this section, it may be a good idea to show a figure with the average percentage of FVC at the Landsat pixel level for the National Reserve during the considered period. Thus, the readers can have better idea about the fraction of vegetation cover of the study area.

Respond 5:

Thanks for your suggestion. We added the FVC area ratio chart in Figure 5 of the revised manuscript. See manuscript Figure 5e for details.

Figure 5. FVC interannual variation and Theil-Sen median trend analysis and Mann-Kendall trend change and area proportion. (a) FVC trend change from 1986 to 2021, (b) MK mutation test, (c) Theil-Sen median trend analysis and Mann-Kendall trend change and area ratio from 1986 to 2009, (d) Theil-Sen median trend analysis and Mann-Kendall trend change and area ratio from 2010 to 2021, (e) the area proportion from 1986 to 2021.

Point 6:

I would not expect two drastic decreases in the temporal behavior of temperature over the period, as shown in Figure 9. I expect much more similar behavior as shown by precipitation. I suggest checking this data with some other data sets available in the internet or, if known, provide some explanation about this temporal signature.

Respond 6:

Thanks for your suggestion. We reprocessed and verified the CRU meteorological data. In addition, we reviewed relevant literature. Drought is accompanied by significant heat. In 2010-2011, the intensity of sudden droughts decreased significantly(Yin et al.,2023), which corresponds to the temperature decrease in the manuscript. He (He et al.,2019) research showed that there were varying degrees of temperature decline in both 1990-1991 and 2010-2011, which corresponds to the temperature drop in the manuscript. At the same time, He (He et al.,2019) et al compared CRU data with meteorological station data and found strong consistency.

Yin X, Wu Y, Zhao W, et al. Spatiotemporal responses of net primary productivity of alpine ecosystems to flash drought: The Qilian Mountains[J]. Journal of Hydrology, 2023, 624: 129865.

He J, Wang N, Chen A, et al. Glacier changes in the Qilian Mountains, Northwest China, between the 1960s and 2015[J]. Water, 2019, 11(3): 623.

Round 2

Reviewer 2 Report

Dear Authors,

Thanks for submitting th revision of the article entitled "Quantitative Analysis of Climate Change and Human Activities on FVC Variations in the Qilian Mountain National Nature Reserve During 1986-2021". I have reviewed the new manuscript and I can acknowledge that the manuscript improved in content and clarity. Most of my major concerns were incorporated and addressed clearly. I have no major concern to disclose.

The manuscript can be accepted.

The English is fine, only minor editing is required.